# COLT: Enhancing Video Large Language Models with Continual Tool Usage

**Yuyang Liu**[*]                                                                    *yuyang.liu@mbzuai.ac.ae*
*Department of Computer Vision*
*University of Mohamed bin Zayed University of Artificial Intelligence*

**Meng Cao**[*]                                                                      *mengcaopku@gmail.com*
*Department of Computer Vision*
*University of Mohamed bin Zayed University of Artificial Intelligence*

**Xinyuan Shi**                                                                      *xinyuan.shi@mbzuai.ac.ae*
*Department of Computer Vision*
*University of Mohamed bin Zayed University of Artificial Intelligence*

**Xiaodan Liang**                                                                    *xiaodan.liang@mbzuai.ac.ae*
*Department of Computer Vision*
*University of Mohamed bin Zayed University of Artificial Intelligence*

**Reviewed on OpenReview:** *https://openreview.net/forum?id=NT9tHHTlXn*

## Abstract

The success of Large Language Models (LLMs) has significantly propelled the research of video understanding. To leverage the strengths of specialist models (*i.e.*, tool) for specific video tasks, recent video LLMs have focused on integrating specialist models as tool usage capabilities into their architectures. Existing methods either prompt closed-source LLMs or employ the instruction tuning paradigm for tool-use finetuning. These methods, however, assume an established repository of *fixed* tools and struggle to generalize to real-world environments where tool data is perpetually evolving and streaming in. To this end, we propose to enhance open-source video LLMs with COntinuaL Tool usage (termed COLT), which automatically acquires tool-use ability in a successive tool stream without suffering "catastrophic forgetting" of the past learned tools. Specifically, our COLT incorporates a learnable tool codebook as a tool-specific memory system. Then, relevant tools are dynamically selected based on the similarity between user instructions and tool features within the codebook. To unleash the tool usage potential of video LLMs, we collect a video-centric tool-use instruction tuning dataset VideoTool. Extensive experiments on both previous video LLM benchmarks and the tool-use-specific VideoTool test split demonstrate the state-of-the-art performance of our proposed COLT.

## 1 Introduction

Large Language Models (LLMs), *e.g.*, GPT (Achiam et al., 2023; Brown et al., 2020), PaLM (Anil et al., 2023), and LLaMA (Touvron et al., 2023a;b), have exhibited remarkable success in understanding user instructions, aligning with human intents, and generating trustworthy responses. Trained on large-scale corpora and equipped with billions of parameters, these models demonstrate impressive generalization abilities across a wide range of natural language tasks such as reasoning, summarization, and question answering. They have fundamentally reshaped the paradigm of natural language understanding by enabling unified instruction-following interfaces that support various downstream applications, from conversational agents

---

[*]Equal contribution.

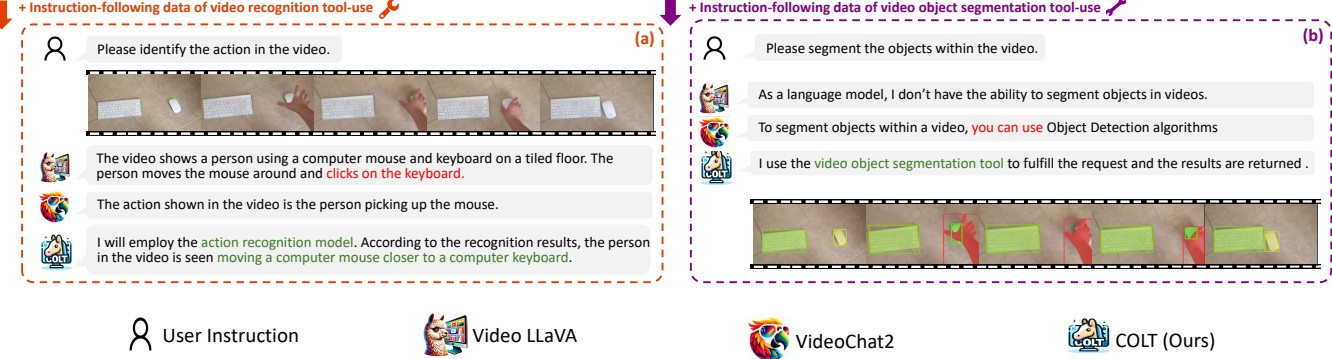

Figure 1: Our proposed COLT continually learns to invoke tools from tool-stream data without catastrophic forgetting. Benefiting from tool usage, COLT (a) is adept at dynamic content understanding and (b) supports flexible generation compared to existing methods Lin et al. (2023a); Li et al. (2023c). The incorrect parts of responses are marked in red.

to coding assistants. Recently, the research focus has been gradually shifting from pure text-based LLMs to multi-modal LLMs such as BLIP-2 (Li et al., 2023a), Flamingo (Alayrac et al., 2022), and MiniGPT-4 (Zhu et al., 2023b), which augment language models with the capability to perceive and reason over multiple modalities such as vision, audio, and video. By integrating visual encoders with pretrained LLMs, these models can interpret images or videos and produce context-aware, grounded responses, thus bridging perception and language understanding. Such a shift extends the role of LLMs from passive text generators to more general perceptual agents, which has aroused significant research interest within the academic community and beyond.

Despite the progress in image-based LLMs, advancing LLMs' capacity to comprehend video data remains a demanding pursuit. Recent representative efforts such as Video-LLaMA (Zhang et al., 2023a) and VideoChat (Li et al., 2023b) illustrate the potential of video LLMs, yet the field is still in its early stage. The overwhelming majority of video LLM methods follow the instruction-tuning paradigm. Building upon open-source LLMs such as Vicuna (Chiang et al., 2023) and LLaMA (Touvron et al., 2023a;b), they entail end-to-end training on instruction-tuning datasets generated by GPT models (Achiam et al., 2023; Brown et al., 2020). These methods, however, demonstrate limited ability to comprehend video actions and present responses with hallucinations (*e.g.*, non-existent "keyboard click" actions). Besides, most of them lack the capability of *any-to-any* generation, *e.g.*, video object segmentation in Figure 1 (b).

To alleviate this, agent-based methods such as Assist-GPT (Gao et al., 2023), ToolLLM (Qin et al., 2023), DoraemonGPT (Yang et al., 2024b), VideoAgent (Fan et al., 2024), and VideoAgent-2 (Wang et al., 2024b) are introduced to advocate the tool-use capabilities of LLMs. More recently, VITAL (Zhang et al., 2025) explores reinforcing tool-augmented video reasoning via multimodal reinforcement learning on long videos. As shown in Figure 2 (a), they mostly bootstrap closed-source LLMs (Achiam et al., 2023; Brown et al., 2020) to decompose the user instructions into more manageable sub-tasks and incorporate external tools to assist. These training-free methods require delicately designed prompts and lengthy context windows for instructions (Liu et al., 2023c). Instead, recent attempts like LLaVA-plus (Liu et al., 2023c) and ToolLLM (Qin et al., 2023) explore developing tool-specific instruction tuning data and elicit the tool-use capabilities of

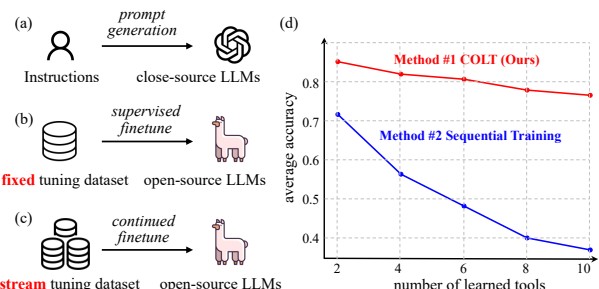

Figure 2: (a) **Agent-based methods** bootstrap closed-source LLMs via delicately designed prompts; (b) Instruction tuning with **fixed** tool-use dataset; (c) Instruction tuning with **stream** tool-use dataset (Ours); (d) **Average tool calling accuracy on VideoTool *vs.* learned tools.** Sequential training denotes training on a sequence of tasks independently.

open-source LLMs (*c.f.* Figure 2 (b)). However, these methods typically assume a pre-set repository containing fixed tools, which is difficult to generalize into the real-world scenario where tool data arrives in a never-ending stream. In such cases, the conventional sequential training, *i.e.*, training on new data with the pre-trained weights on the previous tool data as initialization, leads to significant catastrophic forgetting of prior tool-usage knowledge. For example in Figure 2 (d), the tool-call accuracy of sequential training rapidly decreases as the number of tools learned increases.

In this paper, we propose to enhance open-source video LLMs with **CO**ntinua**L T**ool usage (dubbed as **COLT**). By "continual", we mean that video LLMs possess the inherent capacity to learn and automatically activate tool functionality in a successive tool-stream format. To achieve this, two critical issues arise: 1) *how to build* a tool-specific memory to mitigate catastrophic forgetting of past knowledge? One intuitive idea is to store a few past tool data and replay them with the new tool data for continual training. However, the requisite memory expands proportionally with the increase in the number of tools, and previous data may be unavailable due to privacy constraints. Instead, we set up a *tool codebook* consisting of learnable prompts to store tool-specific information in a more concise manner; 2) *how to exploit* relevant tools based on input user instructions? For tool activation, we compute the cosine similarity between the embedded user instructions and each prompt within the tool codebook. The highest-response prompts are then selected to "instruct" the model to invoke appropriate tools.

Although several video LLM benchmarks (Maaz et al., 2023; Li et al., 2023c) have been proposed, the community still lacks a benchmark for video-centric tool-use instruction tuning. To this end, we collect **VideoTool** to facilitate tool-use capabilities for open-source video LLMs. Specifically, we collect a set of video specialist models and prompt GPT (Achiam et al., 2023) to generate diverse instructions for tool calls. Then the tool execution results are used to form the instruction-following dataset. Under our continual learning setup, COLT can incrementally incorporate newly added tools from a expanding repository, while mitigating forgetting of previously learned ones.

In summary, our contributions are threefold:

- We proposed COLT, a video LLM with continual tool usage. By maintaining a tool codebook, COLT incrementally learns new tools without catastrophic forgetting.
- A tool-using instruction-following dataset VideoTool is introduced to unlock the potential for tool usage within video LLMs.
- Experiments on both existing video LLM benchmarks and the VideoTool test split have manifested the state-of-the-art performance of COLT. For example, on MSRVTT-QA (Chen & Dolan, 2011), our COLT boosts the previous SOTA method (Li et al., 2023c) by 8.2% on the accuracy of zero-shot video-question answering.

## 2 Related Work

**Video LLMs.** Recent successes of LLMs (Achiam et al., 2023; Anil et al., 2023) have shed light on the burgeoning proliferation of video LLMs (Lin et al., 2023a; Maaz et al., 2023; Gao et al., 2023). The mainstream video LLMs follow the instruction-tuning paradigm. Building upon open-source LLMs (Chiang et al., 2023; Touvron et al., 2023a;b), this kind of method adapts the pre-trained video features into LLM understandable representations via multi-layer perception projector (Lin et al., 2023a; Maaz et al., 2023; Li et al., 2023d; Munasinghe et al., 2023; Liu et al., 2024b; Jin et al., 2023; Liu et al., 2023b), Q-former (Zhang et al., 2023a; Li et al., 2023b), or discretization tokenizer (Jin et al., 2024). These methods, however, fail to generalize to broader video understanding tasks, which may require flexible input and output formats (Wu et al., 2023; Zhan et al., 2024). Several methods attempt to achieve this by employing additional functional modules (Jin et al., 2024; Munasinghe et al., 2023) (*e.g.*, grounding (Munasinghe et al., 2023) or diffusion modules (Jin et al., 2024)), which are not flexible enough to adapt to diverse video understanding needs. To this end, another stream of agent-based methods (Qin et al., 2023; Yang et al., 2024b; Fan et al., 2024; Wang et al., 2024b) is proposed, where multi-modal agents mostly bootstrap closed-source LLM (Achiam et al., 2023; Brown et al., 2020) to decompose solution paths and call external tools. This kind of method relies heavily on delicately designed prompts and may fail to acquire the usage of a novel tool. Therefore,

Table 1: **Tool repository** of the proposed VideoTool including single and compositional tools.

| | Tool | Specialist Model |
|---|---|---|
| *single tool* | Action Recognition (AR) | VideoMAE Tong et al. (2022) |
| | Dense Video Caption (DVC) | PDVC Wang et al. (2021) |
| | Temporal Action Localization (TAL) | InternVideo Wang et al. (2022a) |
| | Optical Character Recognition (OCR) | EasyOCR (JaidedAI, 2023) |
| | Automatic Speech Recognition (ASR) | Whisper Radford et al. (2023) |
| | Video Relation Detection (VRD) | VidVRD Shang et al. (2017) |
| | Video Object Segmentation (VOS) | VisTR Wang et al. (2021) |
| | Text-to-Video Generation (T2V) | T2V Khachatryan et al. (2023) |
| *multi.* | AR + ASR | VideoMAE + Whisper |
| | AR + VOS | VideoMAE + VisTR |

how to implement automatic invocation of related tools in video LLMs under a non-stationary tool stream remains to be solved. This paper combines the strengths of the above two methodologies to empower video LLMs with automatic and continual tool-use ability.

**Continual Learning.** Continual learning (Wang et al., 2024a; Lee et al., 2017; McCloskey & Cohen, 1989) refers to the ability to incrementally acquire, update, and accumulate knowledge throughout the model lifetime without catastrophically forgetting previously learned information. Conceptually, four varieties of methodologies are posited. Regularization-based approaches (Kirkpatrick et al., 2017; Li & Hoiem, 2017; Feng et al., 2022; Yang et al., 2024a) strike the balance between the old and new tasks by adding explicit regularization terms. Architecture-based approaches (Yoon et al., 2018; Li et al., 2019; Ke et al., 2020) isolate model parameters for different tasks. Rehearsal-based methods (Bonicelli et al., 2022; Chen & Chang, 2023; Lin et al., 2023b) typically use a memory buffer to store several training samples from previous classes, which are used to approximate and recover old data distributions. Prompt-based methods (Wang et al., 2022c; Smith et al., 2023; Wang et al., 2022b; Li et al., 2024) usually construct task-adaptive prompts and select appropriate prompts during inference. This kind of method is rehearsal-free and thus more computationally efficient. L2P (Wang et al., 2022c) introduces the concept of a prompt pool and selects prompts by a query-key mechanism. To overcome the separate optimization issue of L2P (Wang et al., 2022c), CODA-Prompt (Smith et al., 2023) assembles learnable prompts with input-conditioned weights. DualPrompt (Wang et al., 2022b) and KC-Prompt (Li et al., 2024) set up prompt pools to respectively encode task-invariant and task-specific knowledge.

Borrowing the favorable rehearsal-free merit, our COLT inventively devises a video LLM with the continual tool-use learning capability. We differ from current prompt-based methods (Wang et al., 2022c; Smith et al., 2023; Wang et al., 2022b; Li et al., 2024) in the following two aspects. Firstly, our focus is on the more complex task of multi-modal language generation as opposed to the basic image classification task. Secondly, our COLT employs straight-through (Van Den Oord et al., 2017; Bengio et al., 2013) gradient estimation to mitigate the optimization challenges encountered in previous prompt-based approaches (Wang et al., 2022c; Smith et al., 2023; Wang et al., 2022b; Li et al., 2024).

## 3 Dataset Construction

**Dataset Structure.** Each instance in VideoTool contains two rounds of conversations between `human` and `gpt`. The first round includes human instructions related to the video content and the LLM responses on choosing appropriate tools; The second round contains the execution results of the corresponding tools and the final responses from LLMs. Specifically, we follow (Yao et al., 2022; Yang et al., 2023; Liu et al., 2023c) to unify the response format of GPT into three fields, including `thought`, `action`, and `value` to mimic a human-like task-solving procedure.

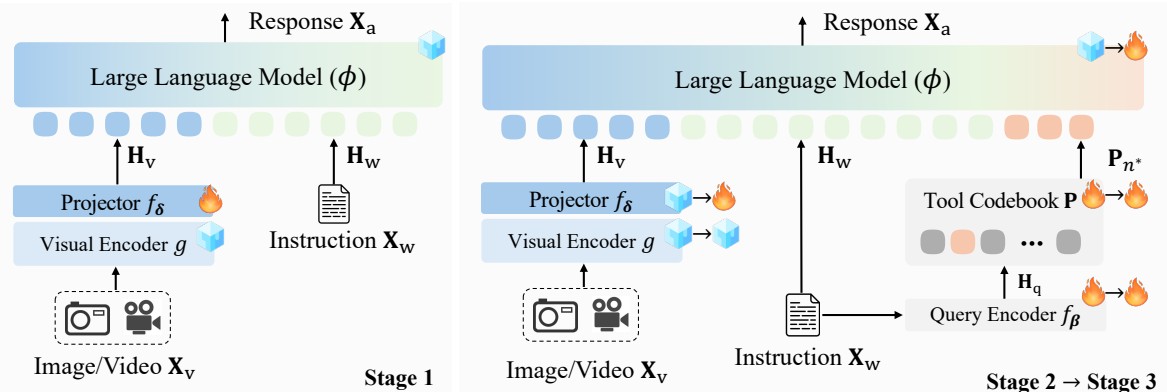

Figure 3: **An overview of COLT.** Stage 1 aligns the visual and textual modalities through the individual training of the linear projector $f_{\boldsymbol{\delta}}$; In stage 2 and stage 3, the prompt within tool codebook $\mathbf{P}$ is adaptively selected according to the cosine similarity with the query feature $\mathbf{H_q}$.

**Dataset Construction.** This dataset is constructed using GPT-3.5-turbo (Achiam et al., 2023) with self-instruction. The involved tools[1] incorporate both video understanding and generation tasks. The prompt and in-context learning cases are available in supplementary materials. As shown in Table 1, we collect ten tools including eight single tool and two compositional tool. The tool list can be easily extended using a similar dataset construction manner. We initially generate 5,000 instruction-following samples for each tool. We conduct both data format checks and manual verification of semantic meanings to filter out error data. After a thorough examination, the dataset is partitioned into distinct training and testing splits, adhering to a proportionality ratio of 9:1. The full statistics of the final VideoTool are summarized in supplementary materials.

## 4 Method

Our COLT learns the continual tool usage from the stream instruction-tuning dataset $\{\mathcal{D}^t\}_{t=1}^T$, each $\mathcal{D}^t$ containing triplets of the visual data $\mathbf{X_v^t}$, user instructions $\mathbf{X_w^t}$, and the response sequence $\mathbf{X_a^t}$. For clarity, we elaborate on the architecture and training strategy for one single dataset $\mathcal{D}^t$ and the superscripts of $\mathbf{X_v^t}$, $\mathbf{X_w^t}$, and $\mathbf{X_a^t}$ are omitted.

### 4.1 Architecture

**Visual & Textual Embedding.** Following (Lin et al., 2023a), we adopt LanguageBind vision encoder (Zhu et al., 2023a) $g(\cdot)$ to extract visual features. Then, a linear projection layer $f_{\boldsymbol{\delta}}(\cdot)$ parameterized by $\boldsymbol{\delta}$ is applied to compress the visual information into an LLM understandable feature space.

$$\mathbf{H_v} = f_{\boldsymbol{\delta}}(g(\mathbf{X_v})), \tag{1}$$

where $\mathbf{H_v} \in \mathbb{R}^{P \times C}$ denotes the visual embeddings, with $P$ and $C$ respectively representing the patch number and the feature dimension. For input user instructions $\mathbf{X_w}$, we adopt the widely used BPE tokenizer (Sennrich et al., 2016) to obtain the textual embeddings $\mathbf{H_w} \in \mathbb{R}^{S \times C}$, where $S$ is the textual token number.

**Tool Codebook & Query Encoder.** As shown in Figure 3, we set up a tool codebook $\mathbf{P}$ as the tool-specific memory for continual tool usage. The codebook consists of $N$ learnable tool prompts, *i.e.*, $\mathbf{P} = \{\mathbf{P}_n\}_{n=1}^N$ with $\mathbf{P}_n \in \mathbb{R}^{C \times 1}$. For each input user instruction $\mathbf{X_q}$, we aim to retrieve relevant tool prompts from the codebook. To achieve this, a *query* encoder $f_{\boldsymbol{\beta}}$ with parameters $\boldsymbol{\beta}$ is firstly employed to encode user instructions $\mathbf{X_q}$ into $\mathbf{H_q} \in \mathbb{R}^{C \times 1}$. Then, we compute the cosine similarities between the query embedding $\mathbf{H_q}$ and each tool prompt $\mathbf{P}_n$, $n \in \{1, 2, \cdots, N\}$. The tool prompts with the top-$K$ highest similarity scores are selected.

$$n^* = \arg \text{topk}_{n \in [1,N]} (\mathbf{P}_n^\top \cdot \mathbf{H_q}), \tag{2}$$

---

[1]In this paper, we use *tool* to represent the general skills and *specialist model* to denote the specific model.

where $n^*$ is the selected tool prompt index. Then each $\mathbf{P}_{n^*}$ is concatenated with the video feature $\mathbf{H_v}$ and user instruction features $\mathbf{H_w}$ and fed into LLMs. The hint contained in the codebook helps alleviate the catastrophic forgetting when facing the tool stream data.

### 4.2 Training

The proposed COLT adopts a three-stage training methodology, with each stage featuring unique trainable parameters $\boldsymbol{\theta}$ and loss functions.

**Stage 1: Video-to-text Alignment.** This stage aims to train the projection layer $f_{\boldsymbol{\delta}}$ parameterized by $\boldsymbol{\delta}$, which acts as a visual tokenizer to align visual signals with pre-trained LLM word embedding. As shown in Figure 3, we freeze all the weights except the projection layer in this stage, *i.e.*, trainable parameters $\boldsymbol{\theta} = \boldsymbol{\delta}$. Given the predicted response sequence $\mathbf{X_a} = \{\mathbf{X_a^1}, \mathbf{X_a^2}, \cdots \mathbf{X_a^L}\}$ of length $L$, we use the vanilla auto-regressive language modeling (LM) loss to supervise this stage of training:

$$\mathcal{L} = \mathcal{L}_{\text{LM}} = -\frac{1}{L}\sum_{i=1}^{L}\log p_{\boldsymbol{\theta}}(\mathbf{X_a^i} \mid \mathbf{H_v}, \mathbf{H_w}, \mathbf{X_a^{<i}}), \tag{3}$$

where $\mathbf{X_a^{<i}}$ is the response tokens before the $i$-th token.

**Stage 2: Tool Codebook Pre-training.** In this stage, we pre-train the tool codebook $\mathbf{P}$ and the query encoder $f_{\boldsymbol{\beta}}$ while keeping the other parts frozen, *i.e.*, the trainable parameters are $\boldsymbol{\theta} = \{\mathbf{P}, \boldsymbol{\beta}\}$. We firstly select the appropriate tool prompt $\mathbf{P}_{n^*}$ via Eq.equation 2. Then $\mathbf{P}_{n^*}$ serves as an additional condition for language modeling as follows:

$$\mathcal{L}'_{\text{LM}} = -\frac{1}{L}\sum_{i=1}^{L}\log p_{\boldsymbol{\theta}}(\mathbf{X_a^i} \mid \mathbf{H_v}, \mathbf{H_w}, \mathbf{P}_{n^*}, \mathbf{X_a^{<i}}), \tag{4}$$

where $\mathbf{X_a^i}$ and $\mathbf{X_a^{<i}}$ is as defined in Eq. equation 3. However, directly optimizing $\mathcal{L}'_{\text{LM}}$ will truncate the gradients w.r.t the query encoder $f_{\boldsymbol{\beta}}$ due to the non-derivable arg topk operation in Eq. equation 2. Therefore, we resort to the straight-through estimator (Van Den Oord et al., 2017; Bengio et al., 2013) to approximate the gradient computation and define the overall loss function as follows:

$$\mathcal{L} = \mathcal{L}'_{\text{LM}} + \lambda_1 \underbrace{\|\text{sg}\,[\mathbf{H_w}] - \mathbf{P}_{n^*}\|_2^2}_{\mathcal{L}_q} + \lambda_2 \underbrace{\|\mathbf{H_w} - \text{sg}[\mathbf{P}_{n^*}]\|_2^2}_{\mathcal{L}_c}, \tag{5}$$

where $\text{sg}[\cdot]$ stands for the stop-gradient operator that acts as an identity in the forward process and has zero partial derivatives during backward propagation. $\mathcal{L}_q$ is the quantisation loss for codebook update by forcing $\mathbf{P}$ towards the user instruction embeddings $\mathbf{H_w}$, $\mathcal{L}_c$ is the commitment loss to prevent unrestricted update of codebook embeddings, $\lambda_1$ and $\lambda_2$ are the balancing weights.

**Stage 3: End-to-end Fine-tuning.** We keep the vision encoder $g(\cdot)$ frozen and finetune remaining parts including the projection layer $f_{\boldsymbol{\delta}}$, tool codebook $\mathbf{P}$, textual encoder $f_{\boldsymbol{\beta}}$ and LLM parameterized by $\boldsymbol{\phi}$, *i.e.*, the trainable parameters are $\boldsymbol{\theta} = \{\boldsymbol{\delta}, \mathbf{P}, \boldsymbol{\beta}, \boldsymbol{\phi}\}$. The training loss is the same as Eq. equation 5.

## 5 Experiments

### 5.1 Experimental Settings

**Training Datasets.** In the first stage, we follow Video LLaVA Lin et al. (2023a) to use LAION-CC-SBU image subset Schuhmann et al. (2021) and the filtered CC3M video dataset Changpinyo et al. (2021) for video-to-text alignment. For the second and third stage, we use the combined dataset of 665K image-text instruction data from LLaVA v1.5 Liu et al. (2023a) and 100K video-text instruction data from Video-ChatGPT Maaz et al. (2023). The tool-use instruction data of our collected VideoTool is introduced into continual training sequentially. To ensure conformity in data formatting, we reformat all the instruction tuning datasets to the `thought-action-value` pattern. Refer to supplementary materials for more details.

**Implementation Details.** For vision encoder $g(\cdot)$, we choose pre-trained LanguageBind Zhu et al. (2023a) with ViT-L/14 Dosovitskiy et al. (2020). The text tokenizer is derived from LLaMA Touvron et al. (2023a)

Table 2: **Comparisons with state-of-the-art methods on zero-shot video-question answering.** "Acc." denotes accuracy (%) and "Score" denotes the relative score from 0 to 5 assigned by GPT Brown et al. (2020). The best performance is in **bold** and the second best is underlined. The backend LLMs include Vicuna-7B Chiang et al. (2023) and LLaMA-7B Touvron et al. (2023a).

| Method | LLM | MSVD-QA | | MSRVTT-QA | | ActivityNet-QA | |
|---|---|---|---|---|---|---|---|
| | | Acc. | Score | Acc. | Score | Acc. | Score |
| Video-LLaMA Zhang et al. (2023a) | Vicuna-7B | 51.6 | 2.5 | 29.6 | 1.8 | 12.4 | 1.1 |
| VideoChat Li et al. (2023b) | Vicuna-7B | 56.3 | 2.8 | 45.0 | 2.5 | 26.5 | 2.2 |
| Video-ChatGPT Maaz et al. (2023) | Vicuna-7B | 64.9 | 3.3 | 49.3 | 2.8 | 35.2 | 2.7 |
| BT-Adapter Liu et al. (2023b) | Vicuna-7B | 67.5 | 3.7 | 57.0 | 3.2 | 45.7 | 3.2 |
| LLaMA-VID Li et al. (2023d) | Vicuna-7B | 69.7 | 3.7 | 57.7 | 3.2 | 47.4 | 3.3 |
| LLaMA-VID Li et al. (2023d) | Vicuna-13B | 70.0 | 3.7 | 58.9 | 3.3 | 47.5 | 3.3 |
| Video-LLaVA Lin et al. (2023a) | Vicuna-7B | 70.7 | 3.9 | 59.2 | 3.5 | 45.3 | 3.3 |
| Chat-UniVi Jin et al. (2023) | Vicuna-7B | 65.0 | 3.6 | 54.6 | 3.1 | 45.8 | 3.2 |
| LLaMA-Adapter Zhang et al. (2023b) | LLaMA-7B | 54.9 | 3.1 | 43.8 | 2.7 | 34.2 | 2.7 |
| VideoChat2 Li et al. (2023c) | Vicuna-7B | 70.0 | 3.9 | 54.1 | 3.3 | 49.1 | 3.3 |
| ST-LLM Liu et al. (2024b) | Vicuna-7B | 74.6 | 3.9 | 63.2 | 3.4 | 50.9 | 3.3 |
| COLT $_{joint}$ (Ours) | Vicuna-7B | **78.2** | **4.2** | **65.1** | **3.6** | **54.7** | **3.8** |
| COLT $_{5 \times 2}$ (Ours) | Vicuna-7B | 75.5 | 3.9 | 63.9 | 3.5 | 52.5 | 3.5 |
| COLT $_{10 \times 1}$ (Ours) | Vicuna-7B | 74.7 | 3.9 | 63.0 | 3.4 | 51.2 | 3.4 |

with a vocabulary size of 32,000, and Vicuna-7B v1.5 Chiang et al. (2023) is employed as the large language model. We uniformly sample 8 frames from each video, and each frame is resized to $224 \times 224$. We set the batch size to 256 for the first stage and 128 for the second and third stages. AdamW optimizer is used with a cosine decay schedule. We set learning rate to $1 \times 10^{-4}$, $1 \times 10^{-4}$, and $1 \times 10^{-5}$ for three stage training, respectively. The balancing weight $\lambda_1$ and $\lambda_2$ in Eq. equation 5 are set to 1 and 0.25. The codebook size $N$ and selected prompt number $K$ are set to 50 and 3, respectively. COLT is trained on 8 NVIDIA A100 GPUs (80 GB memory each), and the full training process takes approximately 10 hours.

**Evaluation Metrics of Tool Continual Learning.** We detail the metrics of average accuracy and average forgetting used in Sec. 5.3.

Let $\alpha_{k,j} \in [0,1]$ denote the tool call accuracy of $j$-th tool after incrementally training on the $k$ tool data ($j \leq k$). The *average accuracy* of a specific tool denotes the overall call accuracy during the incremental learning process. Then the average accuracy at task $k$ is defined as follows.

$$\text{AA}_k = \frac{1}{k} \sum_{j=1}^{k} a_{k,j}. \tag{6}$$

Since average accuracy does not provide any information about the forgetting profile of the continual learning process, *average forgetting* is introduced to bridge the gap. For a particular tool, the forgetting measure is defined as the difference between the maximum tool call accuracy throughout the past learning process and the current tool call accuracy. In particular, the forgetting for the $j$-th tool after incrementally training up to $k$ tool is as follows.

$$f_j^k = \max_{l \in \{1, \cdots, k-1\}} a_{l,j} - a_{k,j}, \quad \forall j < k. \tag{7}$$

The average forgetting of $k$-th tool is computed by normalizing against the number of tools seen previously:

$$\text{AF}_k = \frac{1}{k-1} \sum_{j=1}^{k-1} f_j^k. \tag{8}$$

We report the average accuracy and average forgetting after the last tool learning.

Table 3: **Comparisons (%) with state-of-the-art methods on MVBench.** The tasks include Action Sequence (AS), Action Prediction (AP), Action Antonym (AA), Fine-grained Action (FA), Unexpected Action (UA), Object Existence (OE), Object Interaction (OI), Object Shuffle (OS), Moving Direction (MD), Action Localization (AL), Scene Transition (ST), Action Count (AC), Moving Count (MC), Moving Attribute (MA), State Change (SC), Fine-grained Pose (FP), Character Order (CO), Egocentric Navigation (EN), Episodic Reasoning (ER), Counterfactual Inference (CI), and the average of all 20 metrics (AVG). The best performance is in **bold** and the second best is underlined.

| Model | AVG | AS | AP | AA | FA | UA | OE | OI | OS | MD | AL |
|---|---|---|---|---|---|---|---|---|---|---|---|
| InstructBLIP Dai et al. (2024) | 32.5 | 20.0 | 16.5 | 46.0 | 24.5 | 46.0 | 51.0 | 26.0 | 37.5 | 22.0 | 23.0 |
| LLaVA Liu et al. (2024a) | 36.0 | 28.0 | 39.5 | 63.0 | 30.5 | 39.0 | 53.0 | 41.0 | 41.5 | 23.0 | 20.5 |
| VideoChatGPT Maaz et al. (2023) | 32.7 | 23.5 | 26.0 | 62.0 | 22.5 | 26.5 | 54.0 | 28.0 | 40.0 | 23.0 | 20.0 |
| VideoLLaMA Zhang et al. (2023a) | 34.1 | 27.5 | 25.5 | 51.0 | 29.0 | 39.0 | 48.0 | 40.5 | 38.0 | 22.5 | 22.5 |
| VideoChat Li et al. (2023b) | 35.5 | 33.5 | 26.5 | 56.0 | 33.5 | 40.5 | 53.0 | 40.5 | 30.0 | 25.5 | 27.0 |
| VideoChat2$_{\text{text}}$ Li et al. (2023c) | 34.7 | 24.5 | 27.0 | 49.5 | 27.0 | 38.0 | 53.0 | 28.0 | 40.0 | 25.5 | 27.0 |
| VideoChat2 Li et al. (2023c) | 51.1 | **66.0** | 47.5 | **83.5** | 49.5 | 60.0 | 58.0 | **71.5** | 42.5 | 23.0 | 23.0 |
| GPT-4V OpenAI (2023) | 43.5 | 55.5 | 63.5 | 72.0 | 46.5 | **73.5** | 18.5 | 59.0 | 29.5 | 12.0 | **40.5** |
| COLT $_{\text{joint}}$ (Ours) | **53.4** | 64.0 | 65.0 | 77.0 | 44.5 | 54.5 | **75.5** | 70.0 | 34.0 | **36.5** | 33.0 |
| COLT $_{5\times2}$ (Ours) | 51.8 | 62.0 | 61.0 | 75.0 | 42.5 | 52.5 | 74.5 | 69.0 | 33.5 | 35.0 | 32.0 |
| COLT $_{10\times1}$ (Ours) | 50.6 | 60.0 | 60.5 | 73.5 | 42.0 | 51.5 | 73.5 | 68.5 | 32.5 | 33.0 | 30.0 |

| Model | AVG | ST | AC | MC | MA | SC | FP | CO | EN | ER | CI |
|---|---|---|---|---|---|---|---|---|---|---|---|
| InstructBLIP Dai et al. (2024) | 32.5 | 46.5 | **42.5** | 26.5 | 40.5 | 32.0 | 25.5 | 30.0 | 25.5 | 30.5 | 38.0 |
| LLaVA Liu et al. (2024a) | 36.0 | 45.0 | 34.0 | 20.5 | 38.5 | 47.0 | 25.0 | 36.0 | 27.0 | 26.5 | 42.0 |
| VideoChatGPT Maaz et al. (2023) | 32.7 | 31.0 | 30.5 | 25.5 | 39.5 | **48.5** | 29.0 | 33.0 | 29.5 | 26.0 | 35.5 |
| VideoLLaMA Zhang et al. (2023a) | 34.1 | 43.0 | 34.0 | 22.5 | 32.5 | 45.5 | 32.5 | 40.0 | 30.0 | 21.0 | 37.0 |
| VideoChat Li et al. (2023b) | 35.5 | 48.5 | 35.0 | 20.5 | 42.5 | 46.0 | 26.5 | 41.0 | 23.5 | 23.5 | 36.0 |
| VideoChat2$_{\text{text}}$ Li et al. (2023c) | 34.7 | 38.5 | 41.5 | 27.5 | 32.5 | 46.5 | 26.5 | 36.0 | 33.0 | 32.0 | 40.0 |
| VideoChat2 Li et al. (2023c) | 51.1 | **88.5** | 39.0 | 42.0 | 58.5 | 44.0 | **49.0** | 36.5 | **35.0** | 40.5 | **65.5** |
| GPT-4V OpenAI (2023) | 43.5 | 83.5 | 39.0 | 12.0 | 22.5 | 45.0 | 47.5 | **52.0** | 31.0 | **59.0** | 11.0 |
| COLT $_{\text{joint}}$ (Ours) | **53.4** | 88.0 | 37.5 | **53.5** | **75.5** | 35.5 | 42.5 | 48.0 | 34.5 | 42.0 | 57.5 |
| COLT $_{5\times2}$ (Ours) | 51.8 | 87.0 | 37.0 | 52.5 | 72.0 | 33.0 | 40.5 | 48.0 | 33.5 | 40.5 | 55.0 |
| COLT $_{10\times1}$ (Ours) | 50.6 | 86.0 | 36.0 | 51.5 | 71.5 | 31.5 | 39.5 | 47.5 | 32.0 | 38.0 | 54.0 |

Table 4: **Ablations studies on MVBench.** (a) training strategies; (b) training losses; (c) prompt selection mechanisms: T-based and V-based denote tool prompt selection based on text and visual features, respectively; (d) prompt positions.

| Stage 2 | Stage 3 | AVG | $\mathcal{L}_q$ | $\mathcal{L}_c$ | AVG | T-based | V-based | AVG | Position | AVG |
|---|---|---|---|---|---|---|---|---|---|---|
| ✗ | ✔ | 50.4 | ✗ | ✔ | 37.7 | ✗ | ✔ | 45.3 | *tool-vision-text* | 51.7 |
| ✔ | ✗ | 49.2 | ✔ | ✗ | 40.3 | ✔ | ✗ | **51.8** | *vision-tool-text* | 51.7 |
| ✔ | ✔ | **51.8** | ✔ | ✔ | **51.8** | ✔ | ✔ | 46.4 | *vision-text-tool* | **51.8** |
| | (a) | | | (b) | | | (c) | | (d) | |

## 5.2 Comparisons with Video LLMs

We set up three model variants: **1)** COLT$_{5\times2}$ receives five successive groups of data and each group contains data of two tools; **2)** COLT$_{10\times1}$ is defined similarly; **3)** COLT$_{\text{joint}}$ receives all tool data at once and is trained with all the data collectively. The performance of COLT$_{\text{joint}}$ is regarded as the upper-bound of the continual learning counterpart.

**Evaluation Benchmarks.** Our experiments are carried out on both established video LLM benchmarks and self-built tool-using datasets: 1) **zero-shot video-question answering**. We experiment on commonly used open-ended question-answer datasets: MSVD-QA Chen & Dolan (2011), MSRVTT-QA Xu et al. (2016), and ActivityNet-QA Yu et al. (2019). Following Li et al. (2023b); Lin et al. (2023a), we use GPT-assisted

(a) **Comparisons with SOTA methods on test split of VideoTool.** AR, VRD, and ASR denote action recognition, video relation detection and automatic speech recognition, respectively. "Acc." denotes accuracy (%) and "Score" denotes the GPT-evaluated score.

| Method | AR | | VRD | | ASR | |
|---|---|---|---|---|---|---|
| | Acc. | Score | Acc. | Score | Acc. | Score |
| VideoLLaVA | 47.7 | 2.89 | 6.0 | 0.82 | 1.83 | 0.77 |
| VideoChat2 | 51.4 | 3.13 | 14.0 | 1.36 | 3.60 | 0.87 |
| VideoChat | 29.0 | 2.31 | 4.5 | 0.90 | 2.80 | 1.28 |
| Videoagent | 68.13 | 3.16 | 16.7 | 1.29 | 1.79 | 0.83 |
| VITAL | 52.23 | 3.09 | 15.28 | 1.24 | 1.98 | 0.85 |
| COLT $_{joint}$ | **77.9** | **3.84** | **24.0** | **1.75** | **24.2** | **1.95** |
| COLT $_{5\times2}$ | 74.8 | 3.60 | 21.3 | 1.48 | 22.6 | 1.50 |
| COLT $_{10\times1}$ | 73.5 | 3.48 | 19.6 | 1.27 | 21.9 | 1.48 |

(b) **Comparisons (%) with continual learning methods on VideoTool.** AA and AF denote average accuracy and average forgetting, respectively.

| Method | Five Groups | | Ten Groups | |
|---|---|---|---|---|
| | AA↑ | AF↓ | AA↑ | AF↓ |
| Sequential | 48.6 | 35.3 | 42.3 | 39.7 |
| Rehearsal (10/Tool) | 52.9 | 32.1 | 48.2 | 36.3 |
| Rehearsal (30/Tool) | 55.2 | 30.2 | 49.3 | 35.5 |
| Rehearsal (50/Tool) | 59.3 | 27.4 | 50.2 | 32.7 |
| L2P Wang et al. (2022c) | 72.4 | 8.3 | 65.7 | 12.4 |
| Dual Wang et al. (2022b) | 75.1 | 6.9 | 68.6 | 7.5 |
| CODA Smith et al. (2023) | 76.3 | 7.0 | 71.0 | 7.2 |
| COLT (Ours) | **79.8** | **5.8** | **74.7** | **6.4** |

evaluation to assess the model's capabilities by reporting the accuracy and relative score; 2) **MVBench** Li et al. (2023c). This benchmark consists of 20 demanding video tasks, each comprising 200 samples presented as multiple-choice questions. These tasks offer a thorough and unbiased evaluation of a model's capacity to comprehend videos. We report the choice accuracy as the metric; 3) **VideoTool**. We built this dataset to probe the abilities enabled by tool proficiency. Since most existing video LLMs only support textual outputs, we select three tools (*i.e.*, action recognition, video relation detection, and automatic speech recognition) and compare our COLT to state-of-the-art video LLMs Lin et al. (2023a); Li et al. (2023c;b). The GPT-evaluated accuracy and scores are reported.

**Performance Analysis.** The comparison results on zero-shot video-question answering, MVBench, and VideoTool test split are summarized in Table 2, Table 3, and Table 5a, respectively. We can conclude with the following findings. **1)** Both COLT $_{joint}$ and COLT $_{5\times2}$ demonstrate superior performance compared to preceding state-of-the-art methodologies. For example on MSRVTT-QA (*c.f.* Table 2), COLT $_{joint}$ remarkably surpasses the previous best performing method VideoChat2 Li et al. (2023c) by 8.2% on the metric of accuracy; **2)** On the test set of VideoToolBench (*c.f.* Table 5a), our COLT consistently outperforms prior approaches by a clear margin, including both existing video-VQA models and representative tool-based agents, demonstrating its effectiveness in tool-intensive scenarios; **3)** Even with a rather lengthy learning curve, COLT $_{10\times1}$ still achieves comparable performance with previous SOTA methods; **4)** The performance of COLT $_{10\times1}$ is slightly worse than COLT $_{5\times2}$, which is consistent with the intuition that tool learning becomes increasingly challenging when facing a lengthening tool-chain.

## 5.3 Comparisons with Continual Learning Methods

We additionally compare the proposed COLT with popular continual learning methods on VideoTool test split to demonstrate the life-long tool-usage learning capability of our method.

**Evaluation Metrics.** To evaluate how the system retains tool-using knowledge over time, we adapt the conventional continual learning metric Wang et al. (2024a); Chaudhry et al. (2018) to our tool-using scenario. We set up two metrics: 1) *average accuracy* of a specific tool denotes the overall call accuracy during the incremental learning process; 2) *average forgetting* is defined as the mean difference between the maximum tool call accuracy throughout the past learning process and the current tool call accuracy. A reduction in the value of average forgetting is indicative of enhanced continual learning capability. Refer to the supplementary materials for the formula and detailed explanations.

**Compared Methods.** We set comparison experiments as follows: 1) *Sequential training*: training on new data with the pre-trained weights on previous tool data as initialization; 2) *Rehearsal training*: replay past tool data (*i.e.*, "buffer") and combine them with new data. Buffer size is respectively set to 10/30/50 for

each tool in experiments; 3) Popular continual learning methods including L2P Wang et al. (2022c), Dual Wang et al. (2022b), and CODA Smith et al. (2023).

**Performance Analysis.** We conduct experiments in two settings, *i.e.*, five/ten groups with two/one tools per group. As shown in Table 5b, COLT consistently exhibits superior performance in both average accuracy and forgetting across both settings. For example, under the five-group setting, COLT outperforms CODA Smith et al. (2023) by 3.5% in average accuracy, highlighting its effectiveness in mitigating catastrophic forgetting. Notably, despite being given rehearsal buffers of size 10/30/50 per tool, rehearsal-based methods remain significantly weaker (Table 5b), as they depend on storing and replaying past data and still struggle with interference across heterogeneous tools. In contrast, COLT achieves stronger retention *without* accessing old samples, demonstrating that its codebook-based tool memory provides a more stable and efficient continual-learning mechanism than rehearsal.

### 5.4 Ablation Studies

We conduct extensive ablation studies to provide more insights into our proposed COLT. The experiments are conducted on MVBench Li et al. (2023c) with the continual learning model variant COLT $_{5 \times 2}$.

**Ablation on Training Strategies.** Our COLT adopts a three-stage training pipeline. We conduct ablation studies on the training process by respectively skipping the second and third stages of training. The comparison results on MVBench Li et al. (2023c) in Table 4a demonstrate that both stages are crucial to the final performance, *e.g.*, skipping the second stage leads to a 1.4% drop in average scores.

**Ablation on Training Loss.** We use the straight-through estimator Van Den Oord et al. (2017); Bengio et al. (2013) for the training of stage 2 and stage 3 (*c.f.* Eq. equation 5), which contains the quantisation loss $\mathcal{L}_q$ and commitment loss $\mathcal{L}_c$ to enables the mutual updates between the selected tool prompts and query features. The ablation results in Table 4b underscore the significance of both quantisation loss $\mathcal{L}_q$ and commitment loss $\mathcal{L}_c$.

**Ablation on Prompt Selection.** Recall that we select the most matched tool prompts based on the similarity between the textual instruction features and codebook (*i.e.*, *text-codebook*). Here we ablate on the prompt selection based on the *vision-codebook* similarity or the average similarities of both. The results are listed in Table 4c, which showcases that the *text-codebook* similarity is more reliable. That could be attributed to the fact it is easier for video LLMs to decide whether to invoke or which specific tool to invoke from the user instructions instead of the input visual data.

**Ablation on Tool Prompt Position.** We ablate on the insertion position of the tool prompt. We've listed three options, including *tool-vision-text*, *vision-tool-text*, and *vision-text-tool*, demonstrating placing the tool feature before/between/after the vision and text features, respectively. The findings in Table 4d indicate that the performance is insensitive with regard to the tool prompt position.

**Ablation on hyper-parameters.** We conduct ablation studies on the codebook size $N$ and the number of selected prompts $K$. The comparison results are listed in Figure 4ii. As shown, the average performance is positively correlated with the codebook size $N$ and reaches saturation at $T = 50$. The optimal results are achieved when setting $K = 3$.

**Visualizations of performance *vs.* continually introduced tasks.** To intuitively show the impact of incrementally introducing tools, we show the phased zero-shot video-question answering performance under different training strategies. Specifically, we report the accuracy score on MSVD-QA dataset in Figure 4i. As shown, equipping video LLMs with dense video caption tools leads to the significant performance boost for all four training strategies. Besides, our proposed COLT shows better learning *stability*, *i.e.*, less forgetfulness when faced with new tool data.

## 6 Limitations

While COLT demonstrates strong continual tool-use capabilities, several limitations remain. First, VideoTool is primarily GPT-generated, which may introduce distribution biases and limit the dataset's real-world

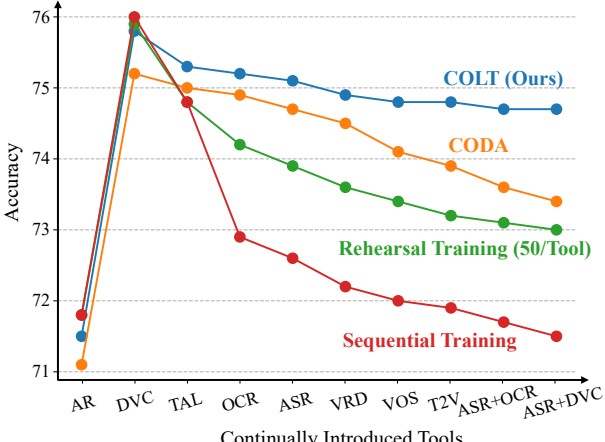

| $N$ | 20 | 30 | 40 | 50 | 60 |
|---|---|---|---|---|---|
| AVG | 48.4 | 50.9 | 51.4 | **51.8** | 51.8 |

| $K$ | 1 | 2 | 3 | 4 | 5 |
|---|---|---|---|---|---|
| AVG | 50.8 | 51.6 | **51.8** | 51.3 | 51.3 |

(i) **Accuracy of zero-shot video-question answering on MSVD *vs*. continually introduced tasks.** The specific tool name for each abbreviation is available in Table 1.

(ii) **Ablations on hyper-parameters** including the codebook size $N$ and the number of selected prompt $K$.

diversity compared with fully human-curated corpora. Second, the current dataset focuses mainly on single-tool and simple multi-tool compositions, which do not fully capture the complexity, interdependence, and noise present in real-world tool ecosystems, where tool chains may be substantially longer and more intricate. Future work includes extending VideoTool to more diverse and human-in-the-loop scenarios, scaling COLT to larger and more complex tool vocabularies, and exploring reinforcement learning or other adaptive strategies to support dynamic and automatic tool composition.

## 7 Conclusions

In this work, we present COLT, which continually learns new tool-using knowledge in a data-stream scenario for general-purpose video understanding. To achieve this, we propose a learnable tool codebook where the specific tool is retrieved and activated according to the similarities with the user instructions. Due to the absence of a video-centric tool-using instruction-tuning dataset within the community, we devised VideoTool to address this deficiency and foster the exploration of tool-using capacities of video LLMs. Experimental results indicate that our proposed COLT adeptly invokes the necessary tools with precision, thereby achieving state-of-the-art performance on widely used benchmarks and the proposed VideoTool test split.

### Broader Impact Statement

The ability of our COLT to continually learn new tool usage without catastrophic forgetting may support the development of *personalized* video LLMs. Data within specific domains or pertaining to particular users can be incrementally fine-tuned, leading to an ever-evolving and personalized intelligent assistant. However, there also exists the risk that the model could be exploited for malicious purposes. Besides, it may raise privacy concerns, especially if it is deployed in surveillance systems or social media platforms.

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

# A    Appendix

The outline of the appendix is as follows:

- Dataset Details
    - Dataset Structure of VideoTool
    - Prompt for Curating VideoTool
    - Instruction Tuning Dataset Reformat
    - Quality Control of VideoTool
- Illustrations of Tool Selection
- Experimental Results
- Qualitative Results

## A.1    Dataset Details

**Dataset Structure of VideoTool.** Each instance in VideoTool contains two rounds of conversation between `human` and `gpt`. The response of `gpt` includes three fields of `thought`, `action` and `value`. One example is available in Table 6.

**Prompt for Curating VideoTool.** We collect VideoTool by self-instruct GPT-3.5-turbo. The prompt and in-context learning cases are shown in Table 7.

**Instruction Tuning Dataset Reformat.** In the second and third stage of COLT training, we reformat the instruction tuning dataset from LLaVA v1.5 (Liu et al., 2023a) and Video-ChatGPT (Maaz et al., 2023) into the `thought-action-value` pattern for the data format conformity. Specifically, the `thought` is generated by GPT indicating that the question can be answered without invoking any tools:

> *The questions can be answered by the information in the context, without need any external tools.*

The `action` is an empty list while `value` is the original response.

**Quality Control of VideoTool:** To ensure the reliability and integrity of the proposed dataset, we conduct quality control from two primary aspects: *data format* checks and of *semantic meaning* checks. The format check entails verifying whether the generated data adheres to a predefined structure, specifically confirming that each entry corresponds to a two-round conversation and that the responses from the GPT model contain essential components such as `thought`/`action`/`value` keys. Furthermore, this phase involves confirming the consistency of tool names used throughout the dataset. Manual checks on semantic meanings are conducted to validate the contextual relevance and accuracy of the generated content, ensuring that it aligns with the intended purpose of the dataset and maintains coherence within the conversations.

Table 6: **Dataset structure of VideoTool.**

| An example of VideoTool dataset |
|---|

*First Round of Conversation*

| Human | Please segment the objects in the video. |
|---|---|
| GPT | \<thoughts\> |
| | Request a process of video object segmentation based on the user's prompt. |
| | \<actions\> |
| | [{'API_name': 'video-object-segmentation', 'API_params': {}}] |
| | \<values\> |
| | Sure thing! I'll initiate a object segmentation model to fulfill your request. Please wait while the processing takes place. |

*Second Round of Conversation*

| Human | You have successfully completed the process of video captioning and object segmentation. |
|---|---|
| GPT | \<thoughts\> |
| | I have successfully completed the process of video object segmentation. Now, it is time to update the user. |
| | \<actions\> |
| | [] |
| | \<value\> |
| | The objects have been segmented. You can now review the results. |

Table 7: **System prompts and in-context learning examples** for the generation of VideoTool.

| Prompts for GPT-3.5-turbo to generate instruction-tuning dataset: |
|---|

You are an AI assistant and you receive a set of conversations in json string format, the content of which is used as data for instruction finetune. Your task is to rephrase the dialogue to generate a new example. Note to give it in standard json format. DO NOT modify the "from" and "actions" parts. Only modify the "value" and "thoughts" part.

| In-context learning examples: |
|---|

{"from": "human", "value": "Transcribe the audio content from the provided video.", "actions": []}, {"from": "gpt", "thoughts": "I need to find an ASR model and initiate the transcription process as per the user's request.", "actions": [{"API_name": "asr", "API_params": {}}], "value": "Certainly! I'll employ an Automatic Speech Recognition (ASR) model. Please wait while the ASR model processes the video"}, {"from": "human", "value": "asr output: You have used an ASR model to transcribe the audio content from the provided video based on my original request.", "actions": []}, {"from": "gpt", "thoughts": "Now that the ASR model has processed the video, I can update the user on the transcription results.", "actions": [], "value": "The ASR transcription you requested has been completed. Here is the transcribed text for your reference."}

## A.2 Illustrations of Tool Selection

Figure 5 shows the full pipeline of our tool selection mechanism.

## A.3 Experimental Results

In Sec. 4.2, we employ an additional CLIP-initialized encoder $f_\beta$ to extract query feature $\mathbf{H_q}$. Then in Eq. equation 2, the appropriate tool prompt is selected based on the cosine similarity with $\mathbf{H_q}$. Here we conduct the ablation study which directly uses the encoded LLM features $\mathbf{H_w}$ for similarity computation. The comparison results in Table 9 demonstrate that the introduced CLIP feature $\mathbf{H_q}$ leads to better performance. In addition, we further analyze how tool-use training and the codebook mechanism affect general video understanding beyond tool-centric benchmarks. As reported in Table 8, removing VideoToolBench data results in performance close to standard video-VQA models, while discarding the codebook causes substantial

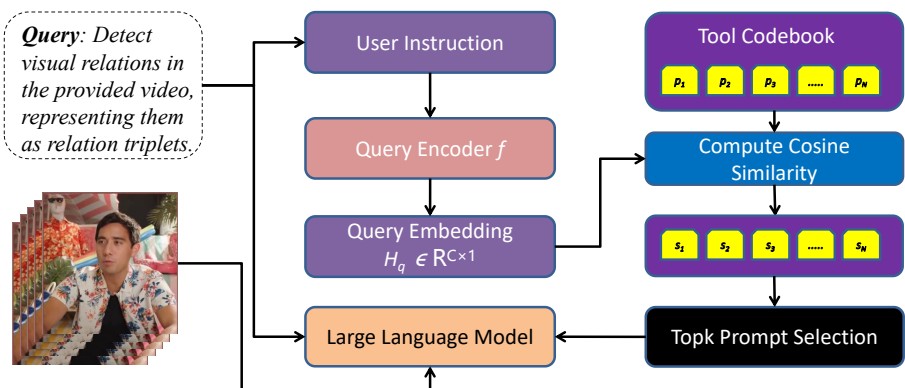

Figure 5: Illustration of the tool selection mechanism based on cosine similarity. Given the query embedding $\mathbf{h}_q$ and a tool codebook $\{\mathbf{p}_1, \ldots, \mathbf{p}_N\}$, we first compute the cosine similarities between $\mathbf{h}_q$ and each tool prompt $\mathbf{p}_i$. The most relevant tools are then selected (e.g., via top-$K$) and concatenated with the visual/text embeddings before being fed into the large language model.

| Method | MSVD-QA | | MSRVTT-QA | | ActivityNet-QA | |
|---|---|---|---|---|---|---|
| | Acc. ↑ | Score ↑ | Acc. ↑ | Score ↑ | Acc. ↑ | Score ↑ |
| COLT (w/o VideoToolBench) | 70.9 | 3.8 | 58.9 | 3.4 | 45.4 | 3.2 |
| COLT (w/o Codebook) | 52.3 | 3.3 | 42.7 | 3.0 | 33.7 | 2.9 |
| COLT-joint | **78.2** | **4.2** | **65.1** | **3.6** | **54.7** | **3.8** |

Table 8: Controlled ablations on standard VQA benchmarks to disentangle the effects of tool-use training (VideoToolBench) and the codebook mechanism.

degradation across all benchmarks. These results suggest that the codebook plays a critical role in preserving and reusing tool-related knowledge, enabling COLT to transfer tool-use behaviors to standard VQA tasks rather than merely memorizing tool traces.

Table 9: **Ablations of the query feature source for tool prompt selection.**

| Query Feature Source | AVG |
|---|---|
| LLM feature $\mathbf{H_w}$ | 47.4 |
| CLIP feature $\mathbf{H_q}$ | **51.8** |

## A.4   Qualitative Results

We present additional qualitative results on MVBench (Li et al., 2023c), zero-shot video-question answering (Chen & Dolan, 2011; Xu et al., 2016; Yu et al., 2019), and VideoTool test split in Figure 6, Figure 7, and Figure 8, respectively. Notably, our COLT precisely captures dynamic video information and generate more reasonable responses.

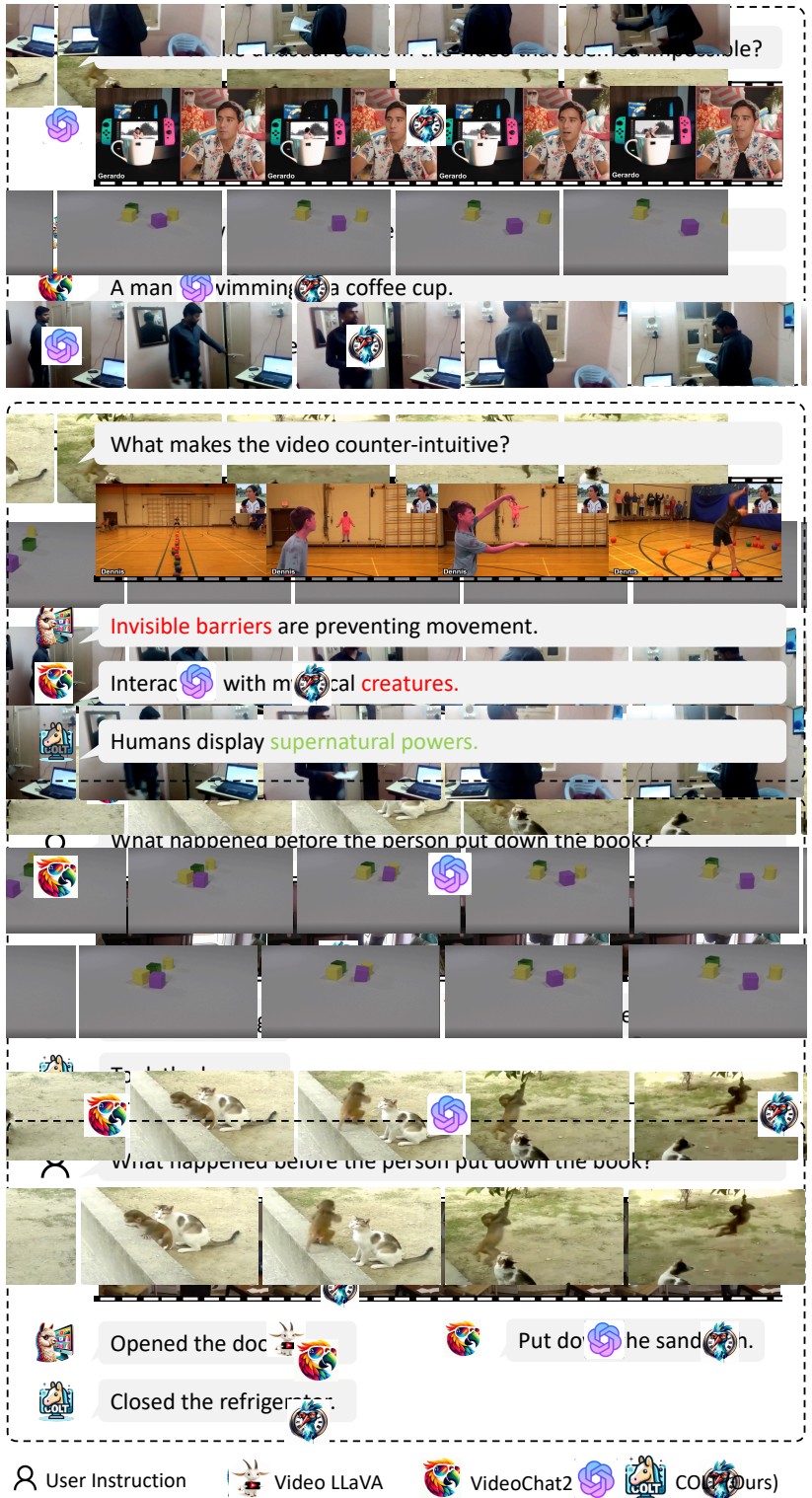

Figure 6: **Qualitative results** on MVBench.

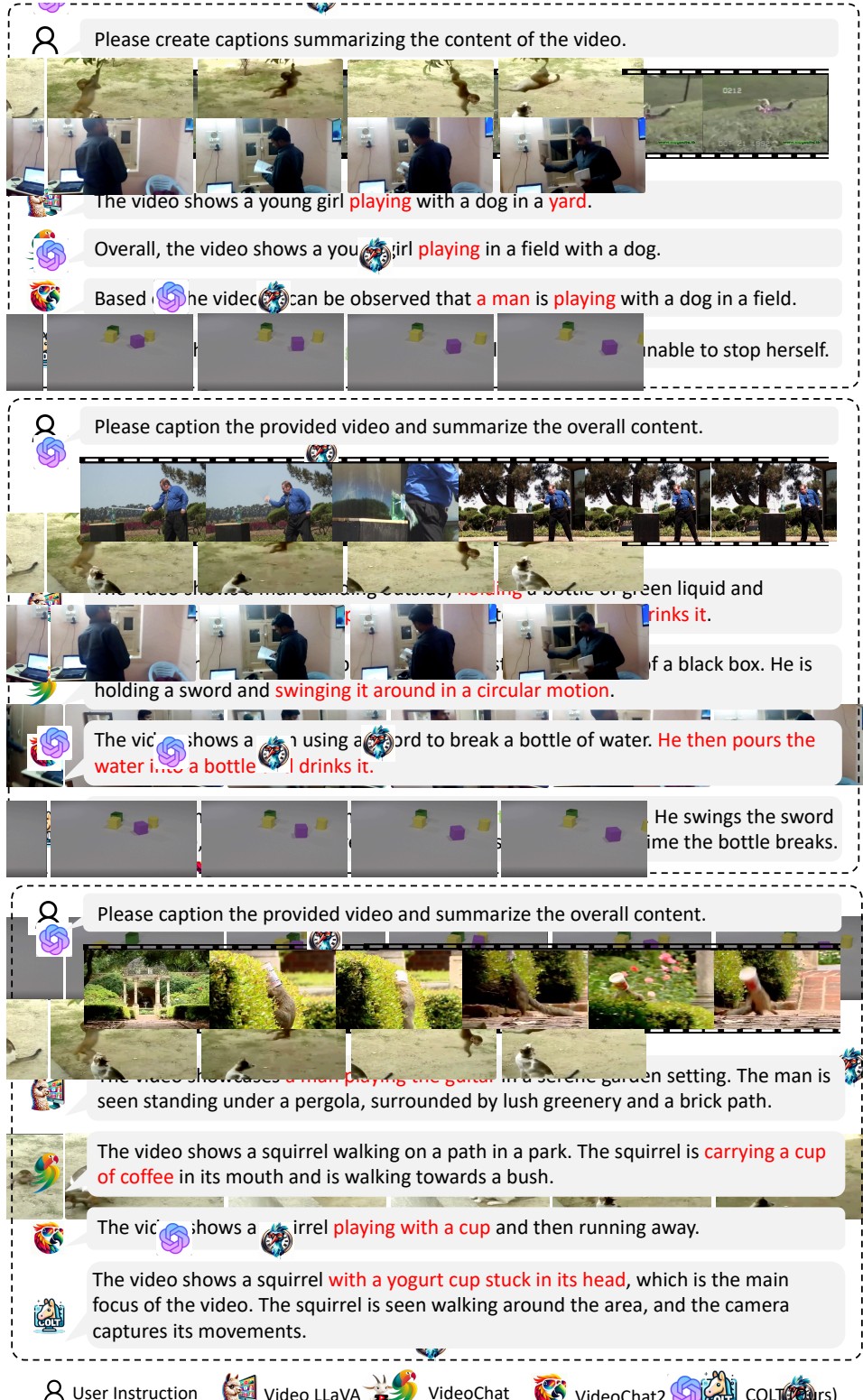

Figure 7: **Qualitative results** on zero-shot video-question answering.

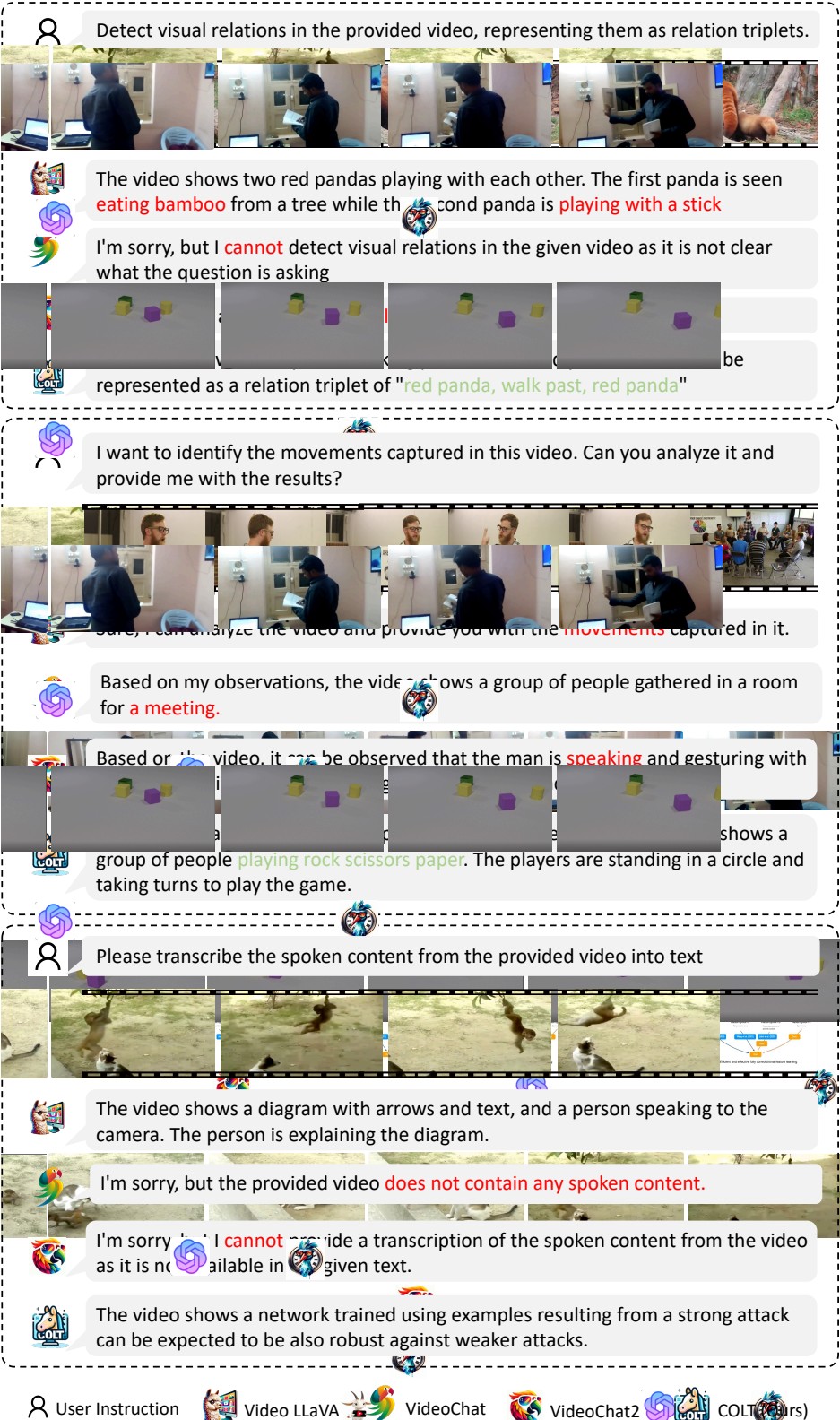

Figure 8: **Qualitative results** on the test split of VideoTool.

