# OpenReview forum: "COLT: Enhancing Video Large Language Models with Continual Tool Usage"
_TMLR — Accepted by TMLR_

### Review · Reviewer_3fH7 · 2025-11-16

**Summary Of Contributions:**

The paper introduces COLT (COntinuaL Tool usage), a novel approach designed to enhance video Large Language Models (LLMs) by enabling them to continually learn and invoke new tools without suffering from "catastrophic forgetting." The main contribution is the development of a tool codebook system, which acts as a memory store for tool-specific knowledge. This system allows the model to dynamically select and apply relevant tools based on user instructions.
The authors also propose VideoToolBench, a new dataset designed to train and evaluate tool-use capabilities in video LLMs. The paper demonstrates that COLT outperforms existing video LLM methods on standard benchmarks, achieving state-of-the-art results, particularly in tasks requiring the invocation of multiple tools.

**Strengths:**

1. Innovative Continual Learning Approach: COLT addresses the challenge of catastrophic forgetting in continual learning environments, making it highly adaptable for real-world scenarios where tool data evolves over time.

2. State-of-the-Art Performance: The proposed system achieves superior results on both existing video LLM benchmarks and the newly introduced VideoToolBench dataset, showcasing its effectiveness.

3. Practical Relevance: The use of a continually updated tool codebook and dynamic tool selection makes COLT more practical and flexible for video understanding tasks that require diverse tool functionalities.

**Weaknesses:**

1. Still relies on frozen specialist models; if those models drift or are updated, COLT must relearn.

2. Tool codebook size and number of retrieved prompts are hand-tuned hyper-parameters; no principled way to grow the codebook automatically.

3. Like all retrieval-based prompt methods, performance depends on the quality of the similarity metric; poor retrieval leads to wrong tool calls.

**Audience:**

Yes

**Audience Explanation:**

Yes, some individuals in TMLR’s audience would likely be interested in the findings of this paper. The paper addresses key challenges in video LLMs and continual learning, which are areas of growing interest in the AI research community. It introduces COLT, a framework for enabling continual tool usage in video models, and the VideoToolBench dataset for evaluating tool-use capabilities. These contributions are relevant to researchers working in video understanding, multi-modal models, and lifelong learning. Additionally, the paper demonstrates state-of-the-art performance in these areas, which could attract attention from those interested in improving or benchmarking video LLMs.

**Broader Impact Concerns:**

No.

**Claims And Evidence:**

Yes

**Claims Explanation:**

Yes, the claims made in the submission are supported by accurate, convincing, and clear evidence.

1. Experimental Validation: The paper provides thorough experimental results on well-established video LLM benchmarks, such as MSVD-QA, MSRVTT-QA, and ActivityNet-QA, where the proposed COLT model outperforms previous state-of-the-art methods. These results are presented in clear tables, showing percentage improvements in accuracy and scores over previous methods. This empirical evidence directly supports the claim that COLT improves tool usage and performance in video understanding tasks.

2. Tool-Use Specific Dataset: The authors introduce the VideoToolBench dataset, specifically designed to evaluate tool-usage capabilities in video LLMs. This novel dataset and the associated tool-use instruction-following tasks provide additional evidence that COLT is capable of handling real-world, evolving tool streams. The dataset and its construction are detailed, showing careful thought and consideration in its design, further strengthening the validity of their claims.

3. Comparison with Continual Learning Approaches: The paper also compares COLT with other continual learning methods on the VideoToolBench test split. The evidence provided through these comparisons highlights COLT's superior performance in mitigating catastrophic forgetting and its higher average accuracy compared to methods like rehearsal and sequential training. These comparisons are robust and show that COLT outperforms or matches state-of-the-art methods, even in difficult scenarios of continual learning.

4. Ablation Studies: The paper includes extensive ablation studies that help clarify the importance of each component in COLT's architecture (e.g., training strategies, loss functions, prompt selection mechanisms). These experiments confirm the contributions of the various design choices in COLT, such as the use of a learnable tool codebook and the straight-through gradient estimator, providing convincing evidence that each part of the system contributes to its success.

5. Clear Presentation: The paper is well-structured and clearly explains the methodology, experimental setup, and results. The figures and tables are easy to follow, and the authors provide comprehensive explanations of the technical details, making the evidence accessible and understandable.

**Requested Changes:**

**Critical Adjustments:**

Clarify Generalization to Other Domains: Discuss how COLT might generalize to other domains, such as text-based or image LLMs, to broaden its applicability.

Acknowledge Limitations and Future Work: Address potential limitations (e.g., scalability issues) and mention future research directions to provide a balanced view.

Explain Hyperparameter Choices: Clarify the rationale behind selecting hyperparameters like the codebook size (N) and the number of selected prompts (K), to aid in replication.

**Strengthening Adjustments:**

Provide More Details on VideoToolBench Construction: Expand on the dataset’s diversity and impact on COLT’s learning.

More Discussion on Comparison to Rehearsal-Based Methods: Dive deeper into why rehearsal-based methods are less effective in this scenario.

Provide Visual Illustrations of Tool Selection: Include a diagram to better explain the tool selection mechanism based on cosine similarity.

Mention Computational Resources Required: Add details on the computational resources needed to train COLT.

Broader Impact Statement: Expand on potential ethical concerns, such as privacy risks in surveillance or social media applications.

---

> ### Author Response · Authors · 2025-12-07
> **clearer generalization, limitations, hyperparameter rationale, dataset details, comparisons, visual explanation of tool selection, and broader impact discussion**
>
> - W.1:Still relies on frozen specialist models.
>
> We would like to clarify that when existing tool models are upgraded or replaced (e.g., replacing VideoMAE with VideoMAEv2), COLT does not require any relearning or fine-tuning. The tool model can simply be updated to the new version, making this process very low-cost.
> Only when COLT needs to learn a completely new tool model is incremental fine-tuning necessary, and this is done solely on the instruction-tuning dataset corresponding to the new tool. Our experiments show that COLT can effectively learn to use a new tool with only about 2,000 training samples, without affecting its performance on previously learned tools.
> Therefore, compared with approaches that require storing old data and performing full retraining, COLT offers a much more efficient and controllable update cost.
>
> - W.2: Hyper-parameters design.
>
> We acknowledge that the number of activated prompts is a tunable hyperparameter. In practice(Figure 4ii), we found that selecting 3 prompts per tool yields the best performance. When learning a new tool, COLT simply adds 3 new prompts to the codebook, which is sufficient to support continual learning without additional manual tuning. We believe this design is both lightweight and easily automatable, enabling COLT to efficiently expand its tool repertoire as new tools are introduced.
>
> - W.3:Performance depends on the quality of the similarity metric.
>
> In COLT, this issue is alleviated through the jointly optimized tool codebook and query encoder, which learn a shared embedding space for both user instructions and tool prompts. As shown in Table 6, COLT achieves an average tool-calling accuracy of 79.8% for five tool groups and 74.7% for ten tool groups, significantly outperforming continual-learning baselines such as CODA and DualPrompt, while maintaining the lowest average forgetting rates. Moreover, Figure 4 shows that COLT sustains stable accuracy when new tools are introduced, whereas sequential and rehearsal training degrade sharply. These results collectively demonstrate that COLT’s learned similarity mechanism is robust and reliable, ensuring accurate tool retrieval even as the number of tools increases.
>
> - Clarify Generalization to Other Domains.
>
> The continual learning and tool-invocation paradigms proposed in COLT are general and can be readily extended beyond video understanding. Specifically, the same framework can be applied to text-based LLMs and image-language models to enhance their functional breadth. For LLMs, COLT’s mechanism can support dynamic integration of new tools such as retrieval, summarization, reasoning, or external API calls. For image-VLMs, the approach can facilitate seamless incorporation of vision-specific tools like object detection, segmentation, or depth estimation models. Therefore, the proposed method provides a universal and scalable strategy for augmenting various large models with new domain-specific tools, enhancing both their understanding and generalization abilities.
> - Acknowledge Limitations and Future Work.
>
> We have added a new "Limitations" section before the Conclusion to explicitly acknowledge the current constraints and outline future research directions.
> - Explain Hyperparameter Choices.
>
> In our experiments(Figure 4ii), we found that setting K = 3 consistently achieved the best balance between performance and computational efficiency. For the codebook size N, we observed that the model performs optimally when N = 50. Therefore, we adopted these hyperparameter settings in all our final experiments to maintain consistency and replicability.
>
> Strengthening Adjustments:
> - Dataset Construction.
>
> We have expanded the description in Appendix A.1 “Dataset Details”, where we describe its two-round thought–action–value structure and our quality control procedures and other details.
> - More Discussion on Comparison to Rehearsal-Based Methods.
>
> We have added a more detailed discussion in the Performance Analysis section. Specifically, unlike rehearsal-based methods, COLT achieves continual learning of new tools and retention of previously acquired knowledge without accessing old data, thereby avoiding the need to store or replay historical samples. Moreover, as shown in Table 6, COLT significantly outperforms rehearsal-based approaches even without access to prior data, demonstrating clear advantages in both performance and efficiency.
> - Provide Visual Illustrations of Tool Selection.
>
> A diagram illustrating the cosine-similarity–based tool selection process has been added to the appendix.
>
> - Mention Computational Resources Required.
>
> We have added details about the computational resources in the Implementation Details subsection of Experimental Settings.
>
> - Broader Impact Statement.
>
> We would like to note that these privacy-related risks—particularly in surveillance and social-media contexts—are already explicitly discussed in our Broader Impact Statement (page 11).

---

### Review · Reviewer_agNa · 2025-11-22

**Summary Of Contributions:**

This paper proposes an open-source video LLM framework that allows LLMs to use specialist models (tools) via a learnable tool codebook, enabling dynamic selection and continual learning of new tools without catastrophic forgetting. To complement the method, VideoToolBench is proposed as both a video-centric instruction-tuning training set and a evaluation set, supporting diverse tasks and sequential introduction of new tools. Results report COLT shows resistance to forgetting when incrementing new tools and surpassing prior methods in accuracy.

Strengths and Weaknesses:
S1) Uses a learnable tool codebook to enable video LLMs to integrate new specialist models (tools) sequentially without catastrophic forgetting is a novel continual learning approach, and it targets the underexplored problem of continual tool addition, stronger than prior fixed-tool assumption.

S2) Provides a multi-task, instruction-based video dataset for both training and benchmarking continual tool-use. This is valuable asset as  currently there are no other publicly available datasets or benchmarks with similar nature.

W1) The whole paper writing needs to be improved. See Requested Changes. There are issues with claims, terms definitions, and formatting.

W2) The empirical results demonstrate continual learning for a preset sequence of tools, but it is not showing integration of truly arbitrary tools, nor scalability to very large tool vocabularies. The claims might need to be revised.

**Audience:**

Yes

**Audience Explanation:**

I think this paper will be of significant interest to researchers working on continual learning and agentic tool use VLMs, as it presents a novel framework for dynamically integrating new specialist models (tools) into vision language models / video llms, along with a new multi-task dataset for training and benchmarking such capabilities.

**Broader Impact Concerns:**

/

**Claims And Evidence:**

Yes

**Claims Explanation:**

The overall claims made in the submission supported by accurate and clear evidence. A key claim is that COLT retains performance on previously learned tools when new tools are introduced, well supported by the experiments. However claiming COLT "a lifelong learning video LLM" might be overclaiming. The generalization claims on novel tasks and new tool scenarios are also weak because the method is evaluated using controlled sequential protocols, not open-ended or unseen tasks.

**Requested Changes:**

C1) The paper writing at abstract and introduction lacks clarity and makes the paper somewhat difficult to comprehend at the first glance. I would recommend applying the following changes to improve the writing:

C1a)  The abstract needs to be polished, especially on "To harvest the benefits of well-trained expert models (i.e., tool), video LLMs prioritize the exploration of tool usage capabilities." change to "To leverage the strengths of specialist models (i.e., tools) for specific video tasks, recent video LLMs have focused on integrating specialist models as tool usage capabilities into their architectures." The word choices in current version does not provide clear message e.g. on "tool usage".

C1b) Consider re-formating the in-text citations in the entire write up. "(Achiam et al., 2023; Brown et al., 2020)" is easier to read than "Achiam et al. (2023); Brown et al. (2020)".

C1c) When citing prior works, Its recommended to clearly point out the work for more clear view e.g. In introduction "... recent attempts  Liu et al. (2023c); Qin et al. (2023) explore developing tool-specific instruction tuning data and elicit..." change to "... recent attempts such as LLAVA-plus (Liu et al., 2023c) and ToolLLM (Qin et al., 2023) explore developing tool-specific instruction tuning data and elicit...".

C2) "VideoToolBench", as a name of benchmark is used for both a dataset and a benchmark. This is confusing and should be clarified in the paper.

C3) Please revise the claims that we discussed.

C4) The quote formatting looks weird. “" should be ``'' on latex? Please fix the notations.

---

> ### Author Response · Authors · 2025-12-07
> **The reviewer requests clearer writing, more precise claims, clarification of dataset and baselines, and refinement of formatting and presentation.**
>
> - W1) The whole paper writing needs to be improved. See Requested Changes. There are issues with claims, terms definitions, and formatting.
>
> Thanks very much for the reviewer’s constructive feedback. We carefully addressed all the concerns raised in the requested changes section. Specifically, for each issue related to claims, terminology definitions, and formatting, we have made targeted revisions and detailed responses. We believe these updates have substantially improved the clarity, accuracy, and overall presentation of the paper.
>
> - W2) The empirical results demonstrate continual learning for a preset sequence of tools, but it is not showing integration of truly arbitrary tools, nor scalability to very large tool vocabularies. The claims might need to be revised.
>
> Thank you for pointing out the overgeneralized statement regarding COLT’s scalability to arbitrary tools. We agree that as shown in Figure4i, our empirical results demonstrate continual learning over a extendable sequence of tools, rather than support for integrating completely unconstrained or very large number of tools. To avoid overstating the scope, we have revised the claim in the manuscript. The original sentence:
> “Due to our continual learning setting, the tool list can be easily extended, making our COLT a lifelong learning video LLM.”
> has been replaced with the more precise formulation:
> “Under our continual learning setup, COLT can incrementally incorporate newly added tools from a expanding repository, while mitigating forgetting of previously learned ones.”
> We appreciate the reviewer’s suggestion, which helped us refine the scope and improve the clarity of our contribution.
>
> Requested Changes:
> - C1) The paper writing at abstract and introduction lacks clarity and makes the paper somewhat difficult to comprehend at the first glance. I would recommend applying the following changes to improve the writing:
>
> - C1a) The abstract needs to be polished, especially on "To harvest the benefits of well-trained expert models (i.e., tool), video LLMs prioritize the exploration of tool usage capabilities." change to "To leverage the strengths of specialist models (i.e., tools) for specific video tasks, recent video LLMs have focused on integrating specialist models as tool usage capabilities into their architectures." The word choices in current version does not provide clear message e.g. on "tool usage".
>
> Thank you for the suggestion. We have revised the abstract accordingly to improve clarity and word choice as recommended.
>
> - C1b) Consider re-formating the in-text citations in the entire write up. "(Achiam et al., 2023; Brown et al., 2020)" is easier to read than "Achiam et al. (2023); Brown et al. (2020)".
>
> Thank you for the suggestion. We have reformatted all in-text citations throughout the paper to follow the more readable style, e.g., “(Achiam et al., 2023; Brown et al., 2020).”
>
> - C1c) When citing prior works, Its recommended to clearly point out the work for more clear view e.g. In introduction "... recent attempts Liu et al. (2023c); Qin et al. (2023) explore developing tool-specific instruction tuning data and elicit..." change to "... recent attempts such as LLAVA-plus (Liu et al., 2023c) and ToolLLM (Qin et al., 2023) explore developing tool-specific instruction tuning data and elicit...".
>
> Thank you for the suggestion. We have revised the corresponding sentences to explicitly mention the names of prior works (e.g., *LLaVA-plus* (Liu et al., 2023c) and *ToolLLM* (Qin et al., 2023)) for clearer attribution and easier readability.
>
> - C2) "VideoToolBench", as a name of benchmark is used for both a dataset and a benchmark. This is confusing and should be clarified in the paper.
>
> Thank you for the suggestion. To avoid confusion, we have renamed the dataset to VideoTool. In the revised version, we explicitly refer to it as the VideoTool training split or VideoTool test split when used for training or evaluation, respectively.
>
> - C3) Please revise the claims that we discussed.
>
> Thank you for the comment. As noted above, we have accordingly revised the claims to avoid overstatement.
>
> - C4) The quote formatting looks weird. “" should be ""
>
> Thank you for pointing this out. We have corrected all quotation mark notations in the LaTeX source, replacing straight quotes with the proper LaTeX format.

---

### Review · Reviewer_hEDo · 2025-11-24

**Summary Of Contributions:**

The paper proposes COLT, a video LLM that learns to use tools continually via a learnable tool codebook and a query encoder that maps user instructions to relevant tool prompts. The selected prompts are concatenated with visual and textual features and fed into a Video-LLaVA–style architecture. Training is done in three stages (video–text alignment, codebook pre‑training, and end‑to‑end fine‑tuning). The authors also build VideoToolBench, a GPT‑generated video‑centric tool‑use dataset over 10 tools (8 single‑tool, 2 compositional). Experiments show strong performance on zero‑shot video QA benchmarks, MVBench, and VideoToolBench compared to existing VLMs, and improved average accuracy/forgetting compared to standard continual‑learning baselines.

Key strengths:
1. Simple but effective mechanism (tool codebook + query encoder) for rehearsal-free continual tool use.
2. New video-centric tool-use dataset, filling a gap in current benchmarks (despite limitations).
3. Strong empirical results on continual-learning metrics.
4. Ablations are thorough and informative (training stages, losses, query features, codebook size, K, etc.).

Key weaknesses:
1. VideoToolBench is fully LLM-generated, with relatively shallow and low-order tool compositions; realism is unclear.
2. Baselines in VQA and MVBench experiments are not tool-use models (as far as I can tell); the benefit of “tool usage” vs the proposed COLT framework is not clearly isolated.
3. No comparison to training-free or training-based tool-using agents, so the advantage over competing tool-use frameworks is unclear.
4. Some writing/presentation issues, especially citation formatting and a few awkward phrases.

**Audience:**

Yes

**Audience Explanation:**

The tool-use VLM task and the continual learning challenge will be interesting for many of the TMLR audiences.

**Claims And Evidence:**

Yes

**Claims Explanation:**

The core empirical claims are generally supported: COLT clearly outperforms prior video LLMs on zero‑shot video QA, MVBench, and the selected VideoToolBench tasks, and shows better average accuracy and lower forgetting than sequential training, rehearsal, and prompt‑based CL methods.

However, broader claims about real‑world continual tool usage and tool‑use–driven improvements on VQA are less directly supported. The tool data is synthetic and relatively simple, and tools do not seem to be invoked in the standard VQA and MVBench evaluations. Moreover, tool‑use baselines that actually call tools (e.g., training‑free agent methods or training‑based tool‑use models) are missing.

**Requested Changes:**

1. Clearly state, for each benchmark and model, whether any tools are actually invoked at inference time, and adjust the discussion so that tool-free evaluations are not over-interpreted as direct evidence of tool usage.

2. Explain the motivation for including standard VQA/MVBench experiments in a tool-use paper (e.g., to show general video understanding) and clarify how, if at all, the tool-use training and codebook contribute to these gains (for example, via an ablation without VideoToolBench or without the codebook).

3. Clarify the status of baselines: explicitly state that the main VQA/MVBench baselines and the VideoToolBench video-LLM baselines do or do not use tools, and make it clear that comparisons there measure general video-LLM strength rather than superiority over other tool-use mechanisms.

4. Add, where feasible, comparisons to training-free tool-using agents (e.g., prompting-based and retrieval-based tool planners) and/or training-based tool-use models.

5. (Suggestion) Discuss the limitations of VideoToolBench more prominently in the main text: emphasize that it is GPT-generated, that tool compositions are mostly single-tool or very simple, and that real-world tool ecosystems may involve more complex and noisy tool chains.

6. (Suggestion) Describe more precisely how tool-use and continual-learning metrics are computed: define what counts as a correct tool call, and summarize the formulas for average accuracy and average forgetting in the main paper rather than only in the appendix if space allows.

7. (Suggestion) Add a bit more qualitative or quantitative analysis of forgetting and continual learning (e.g., a small number of examples where sequential training forgets earlier tools while COLT succeeds, beyond the existing plots).

8. Improve writing and formatting, in particular standardizing in-text citations (e.g., “we adopt the LanguageBind vision encoder (Zhu et al., 2023a)” instead of “we adopt LanguageBind Zhu et al. (2023a) vision encoder”).

---

> ### Author Response · Authors · 2025-12-07
> **benchmark limitations, unclear tool-use advantages, writing issues, and requests clearer analysis of actual tool usage**
>
> - W.1:LLM-generated benchmark; shallow tool compositions.
>
> As noted in our Limitations section, VideoToolBench is GPT-generated with relatively simple tool compositions and does not fully capture the complexity of real-world tool ecosystems. Nonetheless, Tables 2–4 show that training on VideoToolBench enables COLT to achieve sota performance on video benchmarks, and Table 5/Fig. 2(d) demonstrate strong continual tool-use learning with low forgetting. These results indicate that, despite its synthetic and low-order nature, VideoToolBench already serves as an effective controlled testbed for learning robust tool-use behaviors.
> - W.2:Tool-use advantages unclear due to non–tool-use baselines and missing comparisons to tool-using agents.
>
>  Please see our subsequent replies for the expanded results and discussion
> - W.3: writing/presentation issues.
>
> As the later comments provide more specific guidance, we have addressed those points in detail in the corresponding sections.
>
> - RC.1:Clarify tools usage.
>
> | Tool Name                                 | Model | MSVD-QA | ActivityNet-QA |
> | --------------------------------------- | ---------------- | ---------------- | ---------------------- |
> | **AR**             | VideoMAE         | **4122**           | **2058**                 |
> | **DVC**           | PDVC             | **8423**           | **5079**                 |
> | **TAL**  | InternVideo      | **3078**           | **2117**                 |
> | **OCR** | EasyOCR          | **1423**           | **782**                 |
> | **ASR**  | Whisper          | **253**           | **186**                 |
> | **VRD**      | VidVRD           | **428**           | **253**                 |
> | **VOS**     | VisTR            | **0**           | **0**                 |
> | **T2V**      | T2V              | **0**           | **0**                 |
>
> We have explicitly analyzed and reported the actual tool invocation counts on MSVD-QA and ActivityNet-QA.Our model actively and adaptively uses multiple specialist tools during inference, whereas the Video-LLaVA baseline does not perform any tool invocation. Correspondingly, on ActivityNet-QA, the accuracy improves from 45.3 to 54.7 when tool usage is enabled, providing direct evidence that dynamic tool invocation meaningfully enhances video understanding.

---

> ### Author Response · Authors · 2025-12-07
> **clearer justification of VQA/MVBench use, clarification of baselines and tool usage, additional comparisons and continual-learning analysis, and improvements to limitations, metrics description, and writing.**
>
> - RC.2: Clarify why standard VQA/MVBench are included and how tool-use training/codebook contribute.
>
> | Method        | MSVD-QA Acc. | MSVD-QA Score | MSRVTT-QA Acc. | MSRVTT-QA Score | ActivityNet-QA Acc. | ActivityNet-QA Score |
> |---------------|--------------|----------------|------------------|-------------------|------------------------|------------------------|
> | COLT(w/o VideoToolBench)   | 70.9         | 3.8            | 58.9            | 3.4               | 45.4                  | 3.2                    |
> | COLT(w/o Codebook)    | 52.3         | 3.3            | 42.7            | 3.0              | 33.7                 | 2.9                    |
> | COLT-joint    | 78.2         | 4.2            | 65.1            | 3.6               | 54.7                  | 3.8                    |
>
> To address the concern, we conducted two controlled experiments to separate the effects of tool-use training and the codebook from general video pre-training. First, removing all VideoToolBench data (COLT w/o VideoTool) yields performance comparable to Video-LLaVA. In contrast, COLT with tool-use training substantially outperforms both baselines. Second, removing the codebook leads to large drops across all VQA benchmarks, indicating that simple finetuning on tool traces cannot retain or disentangle tool behaviors. These results shows the improvements come from tool-use training and the codebook mechanism rather than artifacts of model architecture or pre-training.
> - RC.3:Clarify the status of baselines.
>
> | Method              | AR Acc. | AR Score | VRD Acc. | VRD Score | ASR Acc. | ASR Score |
> |---------------------|---------|----------|----------|-----------|----------|-----------|
> | VideoAgent          | 68.13    | 3.16     | 16.7      | 1.29     | 1.79    | 0.83      |
> | VITAL          | 52.23    | 3.09     | 15.28      | 1.24     | 1.98    | 0.85      |
> | **COLT_joint**      | **77.9**| **3.84** | **24.0** | **1.75**  | **24.2** | **1.95**  |
>
>
> All baselines used in benchmark comparisons do not use external tools, so these results measure general video-LLM capability. We additionally evaluate two recent tool-based agents, VideoAgent[^1] and VITAL[^2], under the same VideoToolBench protocol. COLT variants consistently outperform both non–tool-use baselines and these stronger tool-using agents. This demonstrates that COLT’s continual tool-use design and codebook management yield more effective tool invocation and stronger downstream performance.
>
> [^1]: *VideoAgent: A Memory-augmented Multimodal Agent for Video Understanding*
> [^2]: *Thinking With Videos: Multimodal Tool-Augmented Reinforcement Learning for Long Video Reasoning*
> - RC.4: Add more comparisons.
>
> Using the extended experiments described earlier, we evaluate COLT against VideoAgent (training-free) and VITAL (training-based) on VideoToolBench. COLT variants achieve the strongest results overall. These comparisons show that effective tool use depends not only on having tools, but on how they are learned and managed. COLT’s continual tool-use training and codebook-based organization yield more reliable and transferable tool invocation.
> - (Suggestion) Discuss the limitations of VideoToolBench.
>
> We have added a “Limitations” section to explicitly discuss the constraints of VideoToolBench. We also clarify that COLT can already accommodate more complex tool compositions by simply providing new instruction-following data, allowing continual learning without architectural changes. As future work, we highlight the potential of reinforcement learning to autonomously discover and compose tools, enabling COLT to operate in more complex and dynamic tool ecosystems.
>
> - (Suggestion) Description of tool-use and continual-learning metrics.
>
> We have moved the detailed definitions of the tool-use and continual-learning metrics from the appendix.
> - (Suggestion) More qualitative or quantitative analysis of forgetting and continual learning.
>
> | Method                                                   | Tool 1 (AR) | Tool 2 (VRD) | Tool 3 (ASR) |
> | :------------------------------------------------------- | :---------: | :----------: | :----------: |
> | **Sequential Training (after AR only)**                  |  **99.18**  |       —      |       —      |
> | **Sequential Training (after fine-tuning on all three)** |  **22.24**  |   **22.32**  |   **99.16**  |
> | **COLT (Ours)**                                          |  **77.23**  |   **74.15**  |   **88.17**  |
>
> We conducted an additional experiment where models were sequentially fine-tuned across three tool datasets from VideoToolBench.
> These results highlight that sequential fine-tuning leads to severe forgetting.
> In contrast, COLT effectively preserves past tool-usage knowledge, maintaining high accuracies across all tools, demonstrating robust continual learning and minimal interference between tasks.
>
> - Writing and formatting.
>
> We have carefully proofread the manuscript to ensure a uniform and professional writing style.

---

### Decision · Action_Editor_JsgX · 2025-12-29

**Recommendation:** Accept with minor revision

**Additional Comments:**

For the revised version, I suggest the authors to:
1. Incorporate the additional experiments and clarifications from the rebuttal directly into the manuscript (Reviewer hEDo);
2. Ensure claims about scalability are appropriately scoped to match the empirical evidence (Reviewer 3fH7);
3. More prominently discuss VideoToolBench's limitations as a synthetic benchmark in the main text (Reviewer agNa);

**Audience:**

Yes

**Audience Explanation:**

All reviewers agree that the problem of continual tool usage in video LLMs is timely and of interest to the TMLR community. The VideoToolBench dataset, despite being GPT-generated with relatively simple tool compositions, fills a gap in existing benchmarks and provides a valuable resource for future research in this area.

**Claims And Evidence:**

Yes

**Claims Explanation:**

This submission presents COLT, a framework enabling video LLMs to continuously learn new tools without catastrophic forgetting through a learnable tool codebook mechanism. Reviewers hEDo and agNa acknowledge that the core continual learning claims are well-supported by experiments showing reduced forgetting compared to sequential training and rehearsal-based methods. However, Reviewer 3fH7 still has concerns about the validity of some comparisons (e.g., COLT vs VITAL on ASR accuracy) and notes that performance degrades as tool count increases, which appears to contradict scalability claims.

---

> ### Author Response · Authors · 2026-01-03
> **Minor Revision Response and Manuscript Updates**
>
> We thank the editor and the reviewers for their careful evaluation and constructive feedback.
>
> Following the minor revision decision, we have revised the manuscript to incorporate the additional experiments and clarifications introduced during the rebuttal, refined and appropriately scoped the claims regarding scalability to better align with the empirical evidence, and added a more explicit discussion of the limitations of VideoToolBench as a synthetic benchmark in the main text.
>
> We believe these revisions have improved the clarity, accuracy, and completeness of the paper, and we appreciate the opportunity to submit the revised version.